# Beiging Modulates Inflammatory Adipogenesis in Salt-Treated and *MEK6*–Transfected Adipocytes

**DOI:** 10.3390/cells10051106

**Published:** 2021-05-04

**Authors:** Songjoo Kang, Myoungsook Lee

**Affiliations:** 1Department of Food & Nutrition, Sungshin Women’s University, Seoul 01133, Korea; crl@sungshin.ac.kr; 2Research Institute of Obesity Sciences, Sungshin Women’s University, Seoul 01133, Korea

**Keywords:** obesity, beiging, beige-like adipocyte (BLA), white adipocyte (WAT), *MEK6*, salt

## Abstract

To investigate whether the beiging process changes the interactive effects of salt and *MEK6* gene on inflammatory adipogenesis, the salt treatment (NaCl 50 mM) and *MEK6* transfection of Tg^(+/+)^ cells were performed with white adipocytes (WAT) and beige-like-adipocytes (BLA). BLA induced by T3 were confirmed by UCP-1 expression and the *MEK6* protein was 3.5 times higher in *MEK6* transfected WAT than the control. The adipogenic genes, *PPAR*-*γ and C/EBP*-*α*, were 1.5 times more highly expressed in the salt-treated groups than the non-salt-treated groups, and adipogenesis was greatly increased in Tg^(+/+)^ WAT compared to non-transfected Tg^(−/−)^. The adipogenesis induced by salt treatment and *MEK6* transfection was significantly reduced in BLA. The inflammatory adipocytokines, TNF-α, IL-1β, and IL-6, were increased in the salt-treated Tg^(+/+)^ WAT, but an anti-inflammation biomarker, the adiponectin/leptin ratio, was reduced in Tg^(+/+)^, to tenth of that in Tg^(−/−)^. However, the production of adipocytokines in WAT was strongly weakened in BLA, although a combination of salt and *MEK6* transfection had the most significant effects on inflammation in both WAT and BLA. Oxygen consumption in mitochondria was maximized in salt-treated and *MEK6* transfected WAT, but it was decreased by 50% in BLA. In conclusion, beiging controls the synergistic effects of salt and *MEK6* on adipogenesis, inflammation, and energy expenditure.

## 1. Introduction

Various obesogenic factors, such as hormone disorder, energy-dense diets, low physical activity, and oxidative stress, may induce body fat accumulation [1,2,3]. We previously found that salt (NaCl) treatment dose-dependently increased the expression of adipogenic/lipogenic genes and the production of pro-inflammatory adipocytokines in WAT [4]. Salt-loading diets (>15 g NaCl/day for seven days) also induced hypoxia in human visceral adipose tissues, which is associated with changes in circulating monocyte subsets (CD14++, CD16+) related to inflammation [5]. Several mechanisms have been suggested to prove the causality in salt-induced obesity, such as monitoring signaling related to adipogenesis/lipogenesis, insulin resistance, and inflammatory environments [3,4,5,6].

The major function of brown adipose tissue in newborns and hibernating mammals is to produce heat through thermogenin, uncoupling protein-1 (UCP-1), by non- shivering thermogenesis instead of through ATP production in the mitochondria of WAT [7]. Brown adipose tissue is decreased or disappears during aging, but beige fat of WAT is sporadically generated in WAT in response to cold temperatures or hormones, which is called beiging or browning [8,9]. The mechanisms related to UCP-1 in beiging, or BLA, may be major mechanisms with which to prevent obesity [10,11]. The regulator of UCP-1 expression, p38, plays a role in various process, such as inflammation, apoptosis, growth, proliferation, and carcinogenesis [12]. In a pilot study, mitogen-activated protein kinase kinase 6 (*MAP2K6, MKK6*) or MAPK/ERK kinase 6 (*MEK6*) gene was screened through.

GWAS in obese children with lower resting metabolic rate (RMR) [13]. *MEK6* pathways are associated with adipogenesis/lipogenesis, and the phosphorylation of p38-α is activated by cold temperatures and upstream regulators of *MEK6* [14]. The depression of p38-α and UCP-1 may decrease energy (heat) production and induce fat accumulation; however, MEK6-activated MAPK/ERK/p38 pathways inhibit beiging [15]. We also confirmed that several mechanisms of salt-induced obesity in the WAT involving *MEK6* gene were also involved in energy expenditure in obese children [16,17].

Although the contributors or target factors for BAT thermogenesis are not fully understood, increasing the thermogenic capacity to prevent obesity should be considered. Because the *MEK6* gene inhibits UCP-1′s action in adipocytes, in this study, we planned to identify whether the beiging process changes the interactive effects of salt and *MEK6*-gene-induced inflammation, adipogenesis, and energy expenditure.

## 2. Materials and Methods

### 2.1. Reagents and Materials

Bovine calf serum (Thermofisher, Waltham, MA, USA), fetal bovine serum (FBS, Capricorn Scientific GmbH, Ebsdorfergrund, Germany), Dulbecco’s modified Eagle’s medium, penicillin-streptomycin, trypsin-EDTA (DMEM, Welgene, Deagu, Republic of Korea), insulin and dexamethasone (DEX, Sigma-Aldrich CO., St. Louis County, MN, USA) were included in the media. For cell staining, Oil Red O staining powder (Sigma-Aldrich CO., St. Louis County, MN, USA), BODIPY powder, and Nile red powder (Thermofisher, Waltham, MA, USA) were used. The MTT assay (Duchefa Biochemie, Amsterdam, The Netherland) was used to measure cell proliferation. For browning, DEX, 3-isobutyl-1-methylxanthine (IBMX), triiodothyronine (T_3_), rosiglitazone were purchased from Sigma-Aldrich CO. (St. Louis County, MN, USA). For *MEK6* transfection, Lipofectamine 3000 (Invitrogen, Carlsbad, CA, USA) was used. Antibodies for PPAR-γ, leptin and UCP-1 (#204350, #3583 and #10983, Abcam, Cambridge, UK); C/EBP-α (#600-1438, Novus Biologycals, Okay, OK, USA); vinculin and MEK-6 (#4650 and #5-15808, Invitrogen, St. Louis, CA, USA); ERK, phospho-ERK (p-ERK)and adiponectin (#4695S, #4370Sand #2789S, Cell Signaling, Beverly, MA, USA) and aP2 (#271529, Santa Cruz Biotechnology, Dallas, TX, USA) were used. The ELISA kits detecting proinflammatory cytokines, TNF-α, MCP-1, plasminogen activator inhibitor (PAI), and IL-6 were provided by R&D Systems (Minneapolis, MN, USA) and IL-1β was provided by Biolegend^®^ (San Diego, CA, USA). The MitoXpress oxygen consumption assay kit (Agilent, CA, USA) was used to detect energy consumption. TRIzol reagent (Ambion, Santa Clara, CA, USA) was used to extract total RNA, and protein assay-dye reagent, and Laemmli 4x buffer (BioRad, Hercules, CA, USA) were used. Ponceau S solution (AMRESCO.Inc., Philadelphia, PA, USA), and ECL kit (GE Healthcare, Chicago, IL, USA), and a polymerase chain reaction (PCR) premix (BioNeer, Daejeon, Republic of Korea) were used.

### 2.2. Cell Culture

The mouse embryo 3T3-L1 preadipocytes was purchased from the American Type Culture Collection (ATCC, Manassas, VA, USA). The 3T3-L1 cells were cultured, as previously described [4,18]. 3T3-L1 cells were cultured in the regular DMEM including 10% bovine calf serum(BCS) and 1% penicillin-streptomycin at 37 °C in a 5% CO_2_ incubator for four days. After reaching confluence, the cells were initiated for adipocyte differentiation by incubation in the DMEM (NaCl 75 mM) of growth medium supplemented with 10% FBS, 1% penicillin-streptomycin, 1 uM DEX, 500 μM IBMX, and 10 ug/mL insulin, for two days (Days 0–1), followed by culture in growth medium supplemented with insulin (10 ug/mL). According to the standard protocol, the 3T3-L1 cells are differentiated to white adipocytes on Day 8 and the medium was changed every 2 days.

### 2.3. Beiging and MEK6 Overexpression in 3T3-L1 Cells

For the beiging process, 50 nM triiodotyronine (T_3_) was added into the growth medium for 2 days, and the medium for beiging (10% FBS-DMEM, 10 ug/mL insulin, 0.5 uM IBMX, 50 nM T_3_, and1 uM rosiglitazone) was changed every 2 days for 8 days. BLA was confirmed by UCP-1 protein expression [19]. For the *MEK6* transfection, *MEK6* DNA (NM_011943) mouse tagged ORF Clone (MG204942, Origene, Rockville, MD, USA) was transfected in the competent bacteria, and plasmid DNA was isolated by pDNA purification kit (Cosmogentech, Seoul, Republic of Korea). A classic, highly effective method of Lipofectamine 3000 reagent, was used for the *MEK6* for over-expression in adipocytes [20]. 3T3-L1 cells (3 × 10^5^ cells/well in 6 well-plate) were cultured for 24 h at 37 °C in 5% CO_2_ until they reached confluence. The DNA mixture was made with Lipofectamine 3000 reagents, p3000 reagent, opti-MEM, and DNA solution. The DNA mixture was activated for 15 min at room temperature and then dropped in to the 3T3-L1 cells. The cells were incubated for 3–4 h at 37 °C in 5% CO_2_. After activation, antibiotics free medium was added to the cells; before culturing them under the same conditions. Transfection of *MEK6* was confirmed by the mRNA(real-time PCR) and protein expression (western blotting).

### 2.4. Cell Viability Assay and Sodium Chloride (NaCl; Salt) Treatment

The concentration of salt (NaCl) to use was determined by the MTT assay. Sodium chloride was diluted to 0 mM, 25 mM, 50 mM, 100 mM, 150 mM, and 200 mM with 10% BCS DMEM(Low-salt; NaCl 75 mM) before being administered to the cells for 24 h. After removal of NaCl in the medium, 10% MTT assay reagent was poured onto the 3T3-L1 cells, and the cells were incubated for 2 h at 37 °C in 5% CO_2_. The MTT assay reagent was removed, and then DMSO was added; the plates were then placed on a shaker for 30 min at room temperature. The absorbance at 590nm and 650nm was detected with a plate reader (Thermofisher).

### 2.5. Oil Red O Staining(ORO)and DAPI/Nile Red Staining

The amount of lipid accumulation was confirmed by ORO staining on Day 8. After washing twice with 1× PBS, the cells were fixed in 4% para-formaldehyde for one hour. The fixing solution was then washed away with 1× PBS and the cells were dried thoroughly. 10% ORO solution was used to stain the lipid droplet of 3T3-L1 cells for 20 min, before they were washed with 1× PBS three times. We monitored the stained cells with a micro-scope. After removing the medium and washing with 1× PBS, the cells were fixed in 4% paraformaldehyde for 10 min. The cells were washed three times with 1× PBS to remove the fixing solution. Nile red and BODIPY solution were diluted 1: 2000 and 1:1000 in 1× PBS, and then added to the cells for staining at room temperature for 15 min. After washing three times for 5 min, we stained the cell nuclei with diluted DAPI solution at room temperature for 30 min. After washing away the DAPI solution with 1× PBS three times for 5 min, each sample was wet-mounted and observed under 400× magnification with a confocal laser scanning microscope (ZEISS, Oberkochen, Germany).

### 2.6. Real-Time PCR(RT-PCR) and Western Blotting Analysis

Total RNA was extracted by TRIzol reagent and RNA concentration was measured by Nano-drop 2000 spectrometer (Thermofisher, MA, USA). cDNA was synthesized from mRNA and amplified using a CFX Connect real-time system (Bio-Rad Laboratories). Primers were designed with Primer-BLAST: *MEK6* (forward: 5′-TGGTGGAGA AGATGC GTCACGT-3′, reverse: 5′-GTCACGGTGAATGGACAGTCCA-3′) and glyceraldehyde-3- phosphate dehydrogenase (forward: 5′-CGTGCCGCCTGGAGAAACC-3′, reverse: 5′-TGGAAGAGTGGGAGTTGCTGTTG-3′).

The Bradford assay was used to determine the protein concentrations. Total protein extracts of cells were isolated and separated on 10% SDS-PAGE gel and transferred the membrane, which were then blocked and incubated with primary antibodies. The method for western blotting with 1st and 2nd antibodies for each target proteins was followed by the general protocols. The protein expression was detected by using the ECL system (GE Healthcare, Chicago, IL, USA) and Chemidoc system (BioRad, CA, USA).

### 2.7. ELISA Assay

The ELISA MAX^TM^ set kit for each inflammatory cytokine factor was purchased from Biolegend (San Diego, CA, USA). For the ELISA assay, 1 mL of supernatant was collected on Day 8. The ELISA was performed with the standard method provided by the manufacturer. (Details: The capture antibody was diluted 1:200 in 100 uL; of diluent sol A capture antibody solution, which was added to each well of the 96 well ELISA plate. The plate was incubated at room temperature overnight.)

### 2.8. Oxygen Consumption (Energy Consumption) Assay

To determine the energy expenditure in Tg^(−/−)^ and Tg^(+/+)^ with salt-treated WAT and BLA, oxygen (O_2_) consumption (RFU/h) in the respiratory system of mitochondria in differentiated and live adipocytes was measured by the Mito Xpress pH-Xtra data visualization tool system (Agilent, Santa Clara, CA, USA) [19].The working solution was prepared by diluting the medium with Xpress agent at a 1:9 ratio. After the differentiation, the cells were incubated with the working solution and two drops of HS mineral oil per well to block the O_2_ in flow from the medium. Absorbance was measured at 380 and 650 nm using a fluorescence plate reader (Molecular Devices), and the O_2_ consumption rate was recorded every 17 sec for a total of 100 min. We collected the data at two points during the experiments, such as at the inflection points of the curve (almost at 30 min from the beginning), and at the end points.

### 2.9. Statistical Analysis

All experiments were performed in triplicate and the data are expressed as the mean ± standard deviation (SD). Comparison of two or more groups was performed by Student’s t-test or one-way analysis of variance (ANOVA) with Duncan’s multiple range test. (IBM SPSS Statistics SW, Ver22). Differences were considered to be statistically significant at *p* < 0.05 and were described by superscript letters (a, b, c, d).

## 3. Results

### 3.1. Salt Treatment, and Characteristics of Beiging and MEK6 Transfection

Cell viability (MTT assay) was dose-dependently increased until 50 mM and decreased above 100 mM, 50 mM was determined to be a suitable concentration without toxicity. (Figure 1A) In the pilot study, we found TG accumulation and MAPK/ERK expression were dose-dependently increased until NaCl 100 mM without morphologic damage [4]. After *MEK6* was transfected into WAT, we confirmed that the *MEK6* mRNA in the Tg^(+/+)^ group was increased by 2.5 times, compared to that in the Tg^(−/−)^, and *MEK6* transfected WAT were also differentiated in the Tg^(−/−)^ group. The *MEK6* protein and MEK6-related phospho-ERK/ERK saw 3.5 and 1.5 times increases in Tg^(+/+)^ WAT compared to Tg^(−/−)^. (Figure 1B) Not only *MEK6* expression but also MAPK pathways activation was also confirmed in the *MEK6* transfected cells. The expression of beiging biomarker, UCP-1, in BLA was a double that in WAT, and we found that *MEK6* expression in BLA was 50% lower than WAT. (Figure 1D) We also confirmed that *MEK6* transfection induced the adipogenic factors, PPAR-γ, C/EBP-α, and aP2 proteins by 4-fold, 3-fold, and 1.5-fold, respectively, compared to Tg^(−/−)^ WAT. (Figure 1C,E) Since the adiponectin/leptin ratio was significantly decreased in *MEK6*-overexpressed WAT, *MEK6* transfection may show the same pattern, with the high production of the inflammatory cytokine, leptin or the low production of the anti-inflammatory cytokine, adiponectin. 

### 3.2. Beiging Modulates Salt Treated- and MEK6 Transfected- Adipogenesis

According to the ORO and DAPI/Nile red staining, the cell sizes and numbers of fat cells were increased in both salt-treated Tg^(+/+)^ and Tg^(−/−)^ compared to non-salted cells, but the beiging process reduced salt-induced fat accumulation. (Figure 2A) The adipogenic effect of salt in the Tg^(+/+)^ group was also reduced under the beiging process compared to in Tg^(−/−)^ group. To determine the combined effects of salt and *MEK6* transfection in WAT and BLA, we compared all of the adipogenesis and beiging related factors between the *MEK6* transfected and control cells. In WAT, the protein expression of adipogenesis related factors such as *MEK6*, PPAR-γ, and C/EBP-α was increased in salt-treated cells, and this pattern was strengthened in Tg^(+^^/^^+)^ to a greater extent than Tg^(−/−)^. (Figure 2B–E) In both salt-treated & *MEK6*-transfected WAT, the highly expressed proteins of *MEK6* and PPAR-γ were strongly decreased in BLA by 50% compared to in WAT. *MEK6* and PPAR-γ were increased only by *MEK6* transfection in BLA, but salt did not change them in BLA. However, C/EBP-α protein was significantly depressed in salt treated- and *MEK6*-transfeced BLA, unlike the patterns of *MEK6* and PPAR-γ. (Figure 2B–E) We concluded that both salt and the *MEK6* transfection affected adipogenesis in WAT, but *MEK6* gene, not salt, was the adipogenic factor in BLA.

### 3.3. Beiging Reduces Salt- and MEK6 Gene-Induced Inflammatory Adipocytokines

Based on the ELISA data, we concluded that the production of pro-inflmmatory and adipocyte-derived cytokines, TNF-α and IL-1β, was increased almost 3 to 5-fold by *MEK6* transfection in WAT, and this pattern was further strengthened in the salt-treated Tg^(+/+)^ groups. (Figure 3) Monocyte chemoattractant protein-1 (MCP-1) was not changed in Tg^(+/+)^ compared to Tg^(−/−)^, nor was plasminogen activator inhibitor-1 (PAI-1) or CD14 affected. (data not shown) IL-6 was increased by 4-fold by both salt treatment and *MEK6* transfection compare to in non-salted Tg^(−/−)^, and salt did not change IL-6 expression in Tg^(−/−)^. Since IL-6 is an interleukin that has both a pro-inflmmatory and anti-inflammatory characteristics, it reacted to salt and *MEK6* transfection. However, the levels of IL-1β, TNF-α and IL-6 were decreased by 90%, 50%, and 60%, respectively in salt treated- and *MEK6* transfected BLA compared to the same conditions of WAT, but MCP-1 was only increased in Tg^(+/+)^ BLA compared to WAT. We concluded that *MEK6* might be involved triggering or activating inflammatory metabolism during the differentiation of adipocytes but beiging positively reduced the synergistic effects of salt and *MEK6* on inflammation.

### 3.4. O_2_ Consumption in WAT and BLA with Salt Treatment & MEK6 Transfection

We also confirmed that the beiging biomarker, UCP-1 expression, was not changed in WAT by neither salt or *MEK6* transfection; however, UCP-1 was increased by 2-fold in BLA compared to WAT. UCP-1 expression was significantly decreased in BLA as salt treatment, and *MEK6* transfection, unlike in WAT. (Figure 4A) Since the O_2_ consumption at the end points almost converged in a parabola around 45 to 50 RFU/h, the threshold points of the curve, at which O_2_ consumption suddenly changed, were compared. At the threshold points of the curve of O_2_ consumption intensity, salt treatment decreased O_2_ consumption in WAT compared to non-salt with Tg^(−/−)^. However, this increased with Tg^(+/+)^ was maximized up to three times in the interaction with salt and *MEK6* transfection in WAT. We found that the *MEK6* gene had a stronger effect on the O_2_ consumption than in the control. Without *MEK6* transfection, O_2_ consumption was higher in BLA and salt-treated BLA than the WAT control. However, the highest intensity (RFU/h) in WAT with synergistic effects of salt and *MEK6* transfection was significantly reduced by 50% in BLA. *MEK6* transfection increased O_2_ consumption at threshold points in both WAT and BLA; moreover, it made the time to reach maximal O_2_ consumption shorter in BLA (data not shown). Therfore, beiging increased thermogenesis instead of through ATP production in mitochondria, even though both salt treatment and *MEK6* transfection were involved in significant factors related to anabolism.

## 4. Discussion

This study is the first to show that the beiging process reduced the synergistic effects of salt- and the *MEK6* gene in obesity. With two pilot studies that salt and *MEK6* gene were involved in fat accumulation, we found that beiging (BLA) reduced salt-and *MEK6* transfection-induced adipogenesis, and production of inflammatory adipocytokines [4,13].

MAPK/ERK pathways are associated with expression of genes involved in adipogenesis, such as insulin resistance, oxidative stress and inflammation [4,21,22,23]. MAPKs and ERK-induced fat synthesis was immediately activated to a great extent after PPAR-γ and C/EBP-α were expressed in WAT [24]. In *ERK* KO mice, the sizes, numbers, and total weights of WAT were decreased compared to those in the control because ERK may inhibit the active form of phosphorylated PPAR-γ [21]. However, we focused that obesity results in a chronic low-grade inflammatory state, with increased pro-inflammatory cytokines, TNF-α and IL-6, causing the activation of stress-induced MAPK signaling [25,26]. Chi et al. reported that MAPK is negatively regulated by MAPK phosphatases and pro-inflammatory factors, TNF-α and IL-6, were highly detrimental to MAPK phosphatase deficient mice [27]. Moreover, IL-6 acted as a pro-inflammatory molecule in this study because it resulted in the a same pattern of increasing TNF-α and IL-1β with salt- and *MEK6* transfection. In the pilot study, we found that the crosstalk among the signaling pathways of MAPK/ERK, Akt-mTOR, and the inflammatory adipogenesis could be the possible mechanism of salt-linked obesity, similarly to in other studies [4,5,28,29,30]. Salt-loading diets (>15 g NaCl/day for seven days)led to consistent hypoxia and the suppression of the circulating renin-angiotensin-aldosterone system, as well as monocyte pro- inflammatory activation [5].

To induce browning, T_3_ is one of the regulators of transcription factors in thermogenesis and mitochondrial function, and rosiglitazone may promote the adrenergic increase in UCP-1, and insulin, as a stimulator of glucose uptake into fat cells [10,30,31,32]. Din et al. found the 52 genes that correlated with UCP-1 in BAT that were involved in the causal relationship between lipid metabolism and a carbohydrate-rich meal triggers human BAT thermogenesis [33,34]. *MEK6* was not involved with any of the 52 genes, but *MEK6* expression was reduced by 50% in BLA while UCP-1 showed a two-fold increase in this study. BAT increases energy expenditure, particularly from glucose in vivo model; however, the metabolic energy rate for fat metabolism was lower in BAT than in the WAT [31]. Sidossis et al. showed that subcutaneous WAT adopts a more thermogenic phenotype after prolonged stress in the burned patients, and browning of WAT was associated with increased whole body metabolic rate [35]. Matesanz et al. proved that browning, T_3_-mediated UCP-1 induction, increased energy expenditure in mice lacking *MKK6*, which protects from high fat diet-induced obesity [15]. Since O_2_ consumption was three times higher in Tg^(+/+)^ groups compared to Tg^(−/−)^ groups over time, the energy demands for fat synthesis in either salt- or *MEK6*-induced adipocytes were increased.

The O_2_ consumption in BLA was higher than that in WAT, but O_2_ consumption in adipocytes depends on meal types as well as exposure temperature [33,34]. In pyruvate kinase KO mice, since the substrates in the media to be taken up in WAT and BLA are identical, salt and the *MEK6* gene are considered the regulators of O_2_ consumption or energy expenditure in WAT and BLA. The metabolic rate for thermogenesis in mitochondria was higher in BLA to enhance energy expenditure and prevent fat accumulation compared to the preferential ATP production in WAT [8,36]. We also found that the time taken for maximal O_2_ consumption was delayed in the transfected groups compared to that in the non-transfected group, and it takes time for the connection between *MEK6* and trigger factors, such as glucose transporter 1 or 4, to affect the maximal O_2_ consumption [35].

To address the limitations of this study, we are planning an in vitro study to elucidate whether browning reduces diet-induced adipogenesis and inflammation in *MEK6* overexpressing mice. However, this first report, that beiging has a complementary effect on the interaction between salt- and *MEK6* gene-induced adipogenesis, will contribute to the obesity research. Although we did not elucidate the mechanisms of *MEK6*’s effects on energy balance during browning, we concluded the *MEK6* pathways are a possible mechanism inducing obesity in a person who has high salt intake. With further clinical trials, these preventive effects on obesity might be more substantial in the future.

## Figures and Tables

**Figure 1 cells-10-01106-f001:**
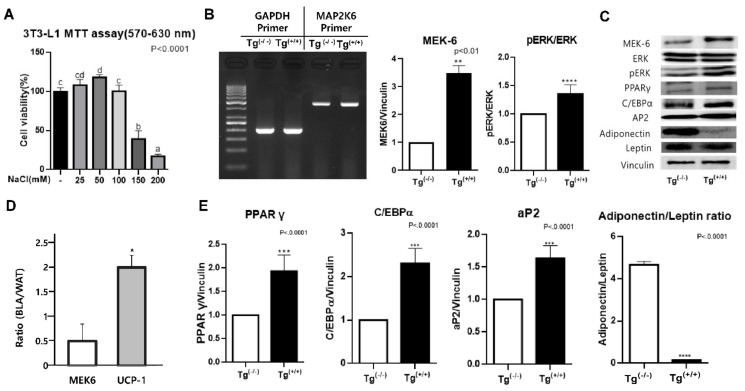
Salt treatment, and characteristics of *MEK6* transfection and beiging. (**A**) Salt treatment level (NaCl 50 mM) was determined by cell proliferation assay. Different superscript letters (a, b, c, d) indicate the significant differences by ANOVA test (*p* < 0.05). (**B**) *MEK6*-transfected adipocytes were confirmed by mRNA and protein levels. (**C**) The expression of proteins related to *MEK6* transfection compared to non-transfection. (**D**) The beiging of WAT(BLA) was confirmed by UCP-1 expression and *MEK6* was depressed in BLA. (**E**) Adipogenesis biomarkers, PPAR-γ, CEBP-α, aP2 and adiponection/leptin ratio, were increased in *MEK6* transfected WAT. Significant differences in individual treatments versus controls were determined by unpaired *t*-test. *; *p* < 0.05, **; *p* < 0.01, ***; *p* < 0.001, ****; *p* < 0.0001.

**Figure 2 cells-10-01106-f002:**
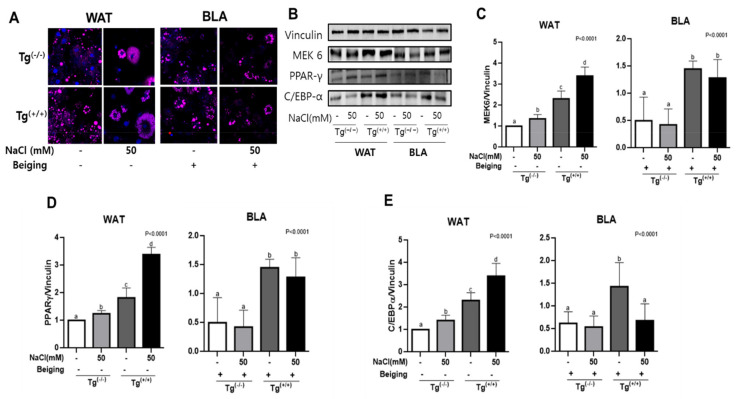
Beiging modulates salt treatment and *MEK6* transfection-induced adipogenesis. Synergistic interaction between salt and *MEK6* transfection affecting fat accumulation (DAPI/Nile red staining; (**A**) according to *MEK6* expression (**B**,**C**), and protein expression related to adipogenesis (**B**–**E**) in WAT, but all effects were weakened by the beiging process. Different superscript letters (a, b, c, d) in individual treatments in WAT or BLA indicate the significant differences from ANOVA test.

**Figure 3 cells-10-01106-f003:**
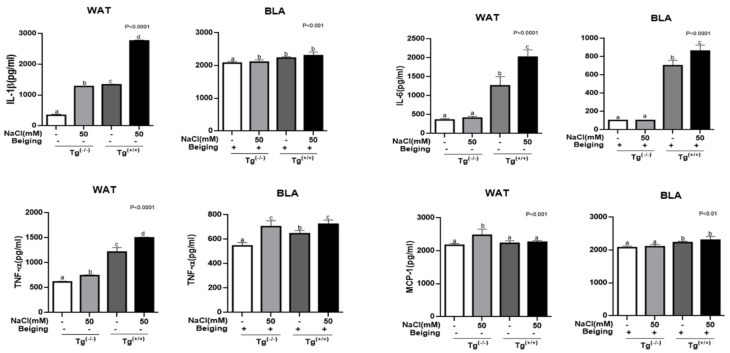
Beiging positively reduced the interactive effects of salt and the *MEK6* gene on the production of inflammatory cytokines, such as TNF-alpha, IL-beta and IL-6, in WAT. Significant differences in individual treatments of WAT or BLA are indicated by ANOVA test with different superscript letters (a, b, c, d).

**Figure 4 cells-10-01106-f004:**
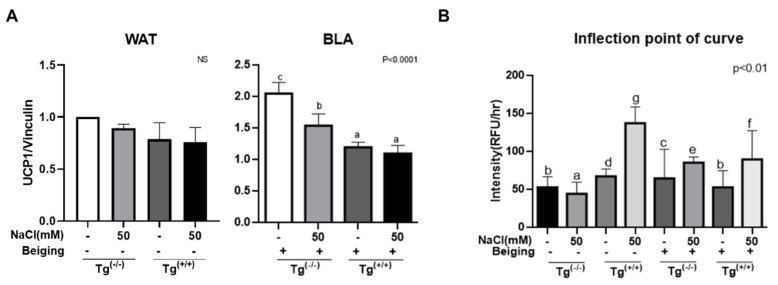
Beiging biomarker, UCP-1, was increased in BLA, but it was decreased by salt treatment and *MEK6* transfection. (**A**) Oxygen consumption intensity (RFU/h) in salt treated-WAT and BLA with Tg^(+/+)^ and without Tg^(+/+)^ were measured at the time of an inflection point of curve in 8th day-culture when cells were completely differentiated and alive. (**B**) Significant differences for individual treatments versus controls are indicated by ANOVA test with different superscript letters letters (a, b, c, d). (NS; non significance).

## Data Availability

Not applicable.

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
