# Peer review of "Beiging Modulates Inflammatory Adipogenesis in Salt-Treated and MEK6–Transfected Adipocytes"

_cells, 2021, doi:10.3390/cells10051106_

Round 1
Reviewer 1 Report
Dear colleagues!
After review of submitted manuscript I have the following comments as a Reviewer assigned by the Editor.
Overall, the study has an original idea and I feel that in current status the report is definitely inappropriately prepared by the Authors.
1) Quality of English and proofing is appalling - one may find >20 typos and style/grammar errors in the first page, so I gave up highlighting them. Many phrases are composed in a very confusing manner, abbreviations and special symbols are used incorrectly. In some sections I have failed to follow the logic and could barely understand the information. The paper must be proofed by an Editor and English should be improved to provide an understandable report to the Reader.
2) Methods are described in a very superficial manner making the study irreproducible. E.g. WB section has not antibody catalogue numbers and some methods are covered in general descriptions of protocols used.
3) In MEK6 overexpression experiments it is not clear what pDNA was used and what was the negative control - "mock" transfection, pDNA backbone w/o MEK6 gene or just untransfected cells (which would be incorrect).
4) In Fig. 1 no radiograms of WB are provided
5) what do small letters (a b c d) in histogram plots are supposed to denote? No information is present.
6) Discussion section is extremely chaotic and despite providing a brief and understandable overview of the study it fails to lead the Reader from hypothesis to conclusion and multiple typos and incorrectly composed sentences make it barely understandable.
I feel that the study has been conducted in a proper way, but current report should be thoroughly prepared for resubmission.
Best, Reviewer.
Author Response
Dear colleagues!
After review of submitted manuscript I have the following comments as a Reviewer assigned by the Editor.
Overall, the study has an original idea and I feel that in current status the report is definitely inappropriately prepared by the Authors.
1) Quality of English and proofing is appalling - one may find >20 typos and style/grammar errors in the first page, so I gave up highlighting them. Many phrases are composed in a very confusing manner, abbreviations and special symbols are used incorrectly. In some sections I have failed to follow the logic and could barely understand the information. The paper must be proofed by an Editor and English should be improved to provide an understandable report to the Reader.
Ans> We received the help from the MDPI-English-editing service (with certificate number #29045), and all misunderstandable parts were reconsidered and revised.
2) Methods are described in a very superficial manner making the study irreproducible. E.g. WB section has not antibody catalogue numbers and some methods are covered in general descriptions of protocols used.
Ans> We inserted catalog numbers of Abs, line 75-79, and methods were revised in detail. Our methods described in this study are definitely reproducible since our lab has been concentrated on obesity works for more than 15 years. However, we added a reference for the BLA model and O2 consumption method that we published in J of Agric Food Chem, 2020. Lipofectamine 3000 reagent for classic transfection methods is highly efficient, visible and reproducible, but we added a classic reference.
3) In MEK6 overexpression experiments it is not clear what pDNA was used and what was the negative control - "mock" transfection, pDNA backbone w/o MEK6 gene or just untransfected cells (which would be incorrect).
Ans> We corrected pDNA information, line 105-108. After the mock transfection (EV w/o MEK6) was compared to the MEK6 transfection of un-transfected cells, we confirmed that the un-transfected cells could be used as the control. Firstly, the reasons were that 3T3-L1 cells did not express lots of MEK6 protein compared to the others, and secondly, the levels of MEK6 mRNA and protein in the transfected cells were effectively higher than that in un-transfected cells. For further studies, when we discussed with Cosmogenetech Co. (http://www.cosmogenetech.com), our Collaborate-Lab, to establish the Stable Cell Line or the animal model of MEK6 over-expression, they confirmed that EV was required for KO transfection to clarify the EV effect on the gene clearance, but it may not be required for the MEK6 over-expression, particularly.
4) In Fig. 1 no radiograms of WB are provided.
Ans> We added in Fig 1-C.
5) what do small letters (a b c d) in histogram plots are supposed to denote? No information is present.
Ans> We corrected them in all legend of figures. Since I found the missing statistics information, I added 2.9 Statistical analysis in the Methods Part, line 159-165.
6) Discussion section is extremely chaotic and despite providing a brief and understandable overview of the study it fails to lead the Reader from hypothesis to conclusion and multiple typos and incorrectly composed sentences make it barely understandable.
Ans> We revised this part as the following principle. We excluded 8 references in the Discussion, and included 2 new references in the Methods.
- First of all, the overview of this study was shortly mentioned in lines 282-285.
- Secondly, MEK6 and salt were involved in obesity with inflammatory obesity mechanism in line 287-309
- Thirdly, I mentioned how browning effect of increasing energy metabolism and also how MEK6 correlated to the browning; lines 310-337
- Lastly, the limitations of this study were in lines 338-346.
Reviewer 2 Report
Beiging modulates inflammatory adipogenesis in salt-treated abd MEK6 transfected adipocytes
The topic of the article is interesting and some improvements can be made to be published.
The abstract can be improved, it is necessary to make a short introduction to the topic and then that to raise the problem.
The article needs to be improved, clarifying from the outset that the salt to which they refer is ClNa, as well as better explain the results, especially histological ones.
In addition, some English adjustments are required.
Author Response
Beiging modulates inflammatory adipogenesis in salt-treated and MEK6 transfected adipocytes
- The topic of the article is interesting and some improvements can be made to be published.
- The abstract can be improved, it is necessary to make a short introduction to the topic and then that to raise the problem.
- In addition, some English adjustments are required.
Ans> The part of Introduction and Discussion to be short and English Writing was edited by MDPI English-editing Services. (with certificate #29045).
- The article needs to be improved, clarifying from the outset that the salt to which they refer is NaCl, as well as better explain the results, especially histological ones.
Ans> I agree to the suggestion about the “salt.” First of all, we planned to prove that high sodium intake may induce obesity in the pilot study, but salt (NaCl) was used instead of “sodium” in vitro. Moreover, since we have to advise people to avoid adding “table salt” in the individual dietary plain, salt means both the NaCl molecule itself and the type of sodium. Since we published salt (NaCl)-induced fat accumulation in WAT (Ref 4), we did not mention here. However, it is better to mention about salt in Methods (line 117-120) and Results (line 178-179) such as “In the pilot study, we found TG accumulation and MAPK/ERK expression were dose-dependently increased until NaCl 100mM without morphologic damage. (Ref 4)”
Round 2
Reviewer 1 Report
Dear colleagues!
All queries have been appropriately addressed by the Authors
Best, Reviewer
This manuscript is a resubmission of an earlier submission. The following is a list of the peer review reports and author responses from that submission.